# Integrative weighted molecular network construction from transcriptomics and genome wide association data to identify shared genetic biomarkers for COPD and lung cancer

Babajan Banaganapalli[1,2]ᴑ*, Bayan Mallah[1,2]ᴑ, Kawthar Saad Alghamdi[3]ᴑ, Walaa F. Albaqami[4], Dalal Sameer Alshaer[1], Nuha Alrayes[2,5], Ramu Elango[1,2], Noor A. Shaik[1,2]*

1 Department of Genetic Medicine, Faculty of Medicine, King Abdulaziz University, Jeddah, Saudi Arabia, 2 Princess Al-Jawhara Al-Brahim Center of Excellence in Research of Hereditary Disorders, King Abdulaziz University, Jeddah, Saudi Arabia, 3 Department of Biology, Faculty of Science, University of Hafr Al Batin, Hafr Al Batin, Saudi Arabia, 4 Department of Science, Prince Sultan Military College of Health Sciences, Dhahran, Saudi Arabia, 5 Department of Medical Laboratory Sciences, Faculty of Applied Medical Sciences, King Abdulaziz University, Jeddah, Saudi Arabia

ᴑ These authors contributed equally to this work.
* bbabajan@kau.edu.sa (BB); nshaik@kau.edu.sa (NAS)

## Abstract

Chronic obstructive pulmonary disease (COPD) is a multifactorial progressive airflow obstruction in the lungs, accounting for high morbidity and mortality across the world. This study aims to identify potential COPD blood-based biomarkers by analyzing the dysregulated gene expression patterns in blood and lung tissues with the help of robust computational approaches. The microarray gene expression datasets from blood (136 COPD and 6 controls) and lung tissues (16 COPD and 19 controls) were analyzed to detect shared differentially expressed genes (DEGs). Then these DEGs were used to construct COPD protein network-clusters and functionally enrich them against gene ontology annotation terms. The hub genes in the COPD network clusters were then queried in GWAS catalog and in several cancer expression databases to explore their pathogenic roles in lung cancers. The comparison of blood and lung tissue datasets revealed 63 shared DEGs. Of these DEGs, 12 COPD hub gene-network clusters (*SREK1*, *TMEM67*, *IRAK2*, *MECOM*, *ASB4*, *C1QTNF2*, *CDC42BPA*, *DPF3*, *DET1*, *CCDC74B*, *KHK*, and *DDX3Y*) connected to dysregulations of protein degradation, inflammatory cytokine production, airway remodeling, and immune cell activity were prioritized with the help of protein interactome and functional enrichment analysis. Interestingly, IRAK2 and MECOM hub genes from these COPD network clusters are known for their involvement in different pulmonary diseases. Additional COPD hub genes like *SREK1*, *TMEM67*, *CDC42BPA*, *DPF3, and ASB4* were identified as prognostic markers in lung cancer, which is reported in 1% of COPD patients. This study identified 12 gene network- clusters as potential blood based genetic biomarkers for COPD diagnosis and prognosis.

**Data Availability Statement:** All relevant data are within the manuscript and its Supporting Information files.

**Funding:** Funding Author: BB Grant Number: G-593-140-1441 Full Name of Funder: Deanship of Scientific Research (DSR), King Abdulaziz University URL: https://dsr.kau.edu.sa/ Role: The funders had no role in study design, data collection and analysis, decision to publish, or preparation of the manuscript.

**Competing interests:** The authors have declared that no competing interests exist.

# 1. Introduction

Chronic obstructive pulmonary disease (COPD) is a progressive airflow obstruction in the lungs which slowly becomes apparent after the 40th or 50th year of age [1]. With a global prevalence of 251 million, COPD disease is currently the fourth leading cause of global deaths and ranked fifth in terms of disease burden [2, 3]. The primary characteristics of the disease are lung inflammation, breathing difficulties, airflow blockage, emphysema, long term cough with mucus, chronic bronchitis, and refractory asthma [4]. Although cigarette smoking is the most well-known significant risk factor for COPD, other factors such as tuberculosis history and environmental exposure to lung irritants (such as indoor air pollutants and occupational dust) are also known to contribute to modifying disease causality and severity [5, 6]. Chronic inflammation is thought to be responsible for pathologic changes such as narrowing of airways in the lungs and destruction of the lung parenchyma, with an underlying role of genetic, epigenetic, and environmental factors [7].

Genetic studies of twins [8], first degree relatives [9] and sporadic COPD cases [10] have all confirmed the role of heritability, which explains at least 30% of the variation in COPD risk. For so long, the genetic basis of COPD has come from Mendelian syndromes, where rare pathogenic variants in *ELN* and *FBLN5* genes cause cutis laxa and *SERPINA1* causes α1-antitrypsin deficiency [11]. Genome-wide association studies have reported the strong association of over 20 genetic loci with COPD and a few additional loci for COPD-related phenotypes like hypoxemia, chronic bronchitis, and emphysema [12]. The molecular basis of COPD, however, could not be fully explained by candidate genetic variants alone, but also by changes in global gene expression. Besides providing an unbiased assessment of thousands of genes in the disease etiology, global gene expression could also potentially help in developing personalized medicine. However, analysis and interpretation of such massive gene expression data is so complex and challenging.

A few studies have attempted to analyze gene expression changes in COPD patients' blood samples in recent years [13–16]. However, the correlation of common gene expression dysregulations between blood and lung tissue samples from COPD patients is not well explored. Recent deployment of advanced statistics and integrative bioinformatics methods, incorporating gene network graphs, unsupervised clustering, and functional annotations of pathways, has provided a new dimension to explore the microarray gene expression datasets to discover the molecular basis of different genetic pathologies [17–19]. Therefore, the objective of this study is to expand our current understanding of COPD pathogenesis and to identify potential genetic biomarkers. By involving a series of comprehensive bioinformatics approaches, this study has identified several gene-network clusters involved in cell communication, inflammation, proliferation, and differentiation processes, are dysregulated in blood and lung tissues of COPD patients. Our findings provide an insight into understanding the mechanisms of COPD and its potential link with lung cancer, besides uncovering genetic markers with potential for disease diagnosis and therapeutic modulation.

# 2. Materials and methods

## 2.1 Microarray gene expression datasets

The NCBI-GEO and EBL-EBI Array Express databases were used to search for COPD gene expression datasets using the keywords like "COPD", "COPD blood", and "COPD tissue". From the output, we selected two COPD gene expression datasets, i.e. GSE8581 and GSE54837 for our study. The first dataset (GSE8581) consists of gene expression data, from 35 lung tissues, which were collected from 16 COPD subjects (with FEV1 < 70% predicted and FEV1/

FVC < 0.7) and 19 controls (with FEV1 > 80% predicted and FEV1/FVC > 0.7), generated on the Affymetrix U133 Plus 2.0 array [20]. The second dataset, GSE54837 includes the expression data generated on GPL570 platform (Affymetrix, Santa Clara, CA, USA) from the blood samples of 136 COPD patients and 6 controls (ex-smokers) [21].

## 2.2 Data preprocessing and analysis

The microarray gene expression data analysis was performed using R/Bioconductor (http://www.R-project.org/). The raw data extracted in.CEL format was normalized into expression values using the Bioconductor-Affy package for the standardization and background correction of the probe data [22]. The limma package was then used to select the statistical significance of the differentially expressed genes between normal and COPD samples by applying the t-test statistical method. The Benjamini-Hochberg method was used to calculate the false discovery rate (FDR) of all the statistically significant genes to enable the removal of false positive ones [24]. The cutoff value for DEGs was set as FDR < 0.01 and |log2 FC| > 1.5. A p-value of less than 0.05 was considered as statistically significant. The expression values of DEGs were divided into up- and down-regulated genes and visualized using the Heatmap online webtool (http://www.heatmapper.ca).

## 2.3 Gene ontology and functional enrichment analyses

Gene Ontology (GO) and KEGG pathway (https://www.genome.jp/kegg/pathway.html) enrichment analysis of DEGs was conducted using STRING database (http://string-db.org). The significant GO terms and pathways were chosen at a threshold of adjusted p< 0.05 and FDR of 0.05. The GO annotation networks were visualized in the Cytoscape network style plugin (http://www.cytoscape.org/).

## 2.4 Construction of protein-protein interaction (PPI) map

The potential PPI networks from the lung and blood DEGs were constructed using Bisogenet, a cytoscape plugin (version 3.4.0). Furthermore, the network clusters from PPI interactions were identified with the help of network analyzer tool. The cut-off value of input nodes and their neighbors was up to a distance of 1 edge. During the creation of PPIM, only protein-protein interactions were selected, excluding protein-DNA interactions and microRNA silencing interactions. Each node represents a gene connected with edges which are physical or functional between the nodes. Therefore, few nodes have a large number of edges while several nodes have low connectivity [23].

## 2.5 Hub gene subnetwork construction

PPIM is considered to be a large-scale network. By following the network biology concepts, the PPIM complex was decomposed into significant subnetwork clusters of Significant Protein Interaction Network (S$^{PIN}$). Based on degree centrality (DC) and betweenness centrality (BC) parameters, several genes were extracted. Each protein captured in the network was incorporated and standardized into Cytoscape 3.2.1 using Network Analyzer to calculate local degree centrality (DC) and global betweenness centrality (BC) parameters of the network [24].

## 2.6 Genome wide association study analysis

The hub genes from the above gene-network clusters were searched in the GWAS catalog database (https://www.ebi.ac.uk/gwas/) to check their association with COPD risk. Variant details like reference and alternate alleles, population frequency, genome wide association value (P-

value of $<5 \times 10^{-8}$), reported trait, and accession number of the study were collected. We have also used another genotype-phenotype association database, PhenoScanner V2 (http://www. phenoscanner.medschl.cam.ac.uk/) to cross reference the association of hub genes with COPD risk. Each hub gene name was searched in the database generated tables, which contain trait specific associations of each gene and genome wide association values for its variants (P-value of $<5 \times 10^{-8}$).

### 2.7 Lung cancer expression analysis

We used three different databases to investigate the expression status of the COPD-hub genes in lung cancer tissues: Gene Expression Profiling Interactive Analysis (GEPIA2), Gene Expression Database of Normal and Tumor Tissues (GENT2), and Human Protein Atlas (HPA). Gene Expression Profiling Interactive Analysis (GEPIA2) (http://gepia2.cancer-pku.cn) was used to provide tumor/normal differential expression analysis. The signature score of hub genes is calculated by mean value of log2 (TPM + 1). The |Log2FC| of 1 and an expression value cutoff of 0.01 (p-value) were determined in Lung Adenocarcinoma (LUAD) and Lung Squamous Carcinoma (LUSC) tissues. The Gene Expression Database of Normal and Tumor Tissues GENT2 (http://gent2.appex.kr/gent2/) platform was used to explore the gene expression patterns across normal and tumor tissues generated from public gene expression data sets. The survival rate status of hub genes in lung cancer and its histological subtypes (adenocarcinomas, large and squamous) represented by Kaplan Meier plots at 95% confidence intervals (CI) and computed log rank *p*-value was determined. The human protein atlas (https://www.proteinatlas.org/) database was used to explore the expression status of each hub gene in human non-malignant and lung cancer tissues. This database takes the query gene or protein name and provides the information about that candidate protein expression based on the primary antibody staining data with a series of immunohistochemistry images of the corresponding clinical specimens.

## 3. Results

### 3.1 Differently expressed gene (DEGs) identification

A total of 54,675 probes were expressed in both datasets. In the human lung tissue dataset (E-GEOD-8581), 678 DEGs including 247 up- and 431 down-regulated genes were identified, whereas, blood dataset (GSE54837) showed the differential expression of 724 DEGs including 499 up- and 225 down-regulated genes. Comparison of both datasets revealed the shared expression of 63 genes (Fig 1A). The expression level of DEGs of COPD patient samples (both tissue and blood) is shown in the form of heatmaps and volcano plots (Fig 1B and 1C).

### 3.2 PPI network analysis and significant genes clusters

Bisogenet, a Cytoscape plugin analysis of DEGs from both datasets generated a complex PPIM network of 1072 nodes (genes) and 20079 edges (interactions). The average edge-node ratio was 18.73 (S1 Fig). In the context of the PPIM network, protein interactions within the same group of clusters are assumed to have similar functions to the less interconnected regions or different cluster groups. Therefore, the Network Analyzer plugin was applied to find significant hub genes with the highest degree of centrality. A total of 12 significant genes and clusters with a degree of centrality of >17 were identified from network analysis (Fig 2; S2 Fig) and chosen as hub proteins (Table 1).

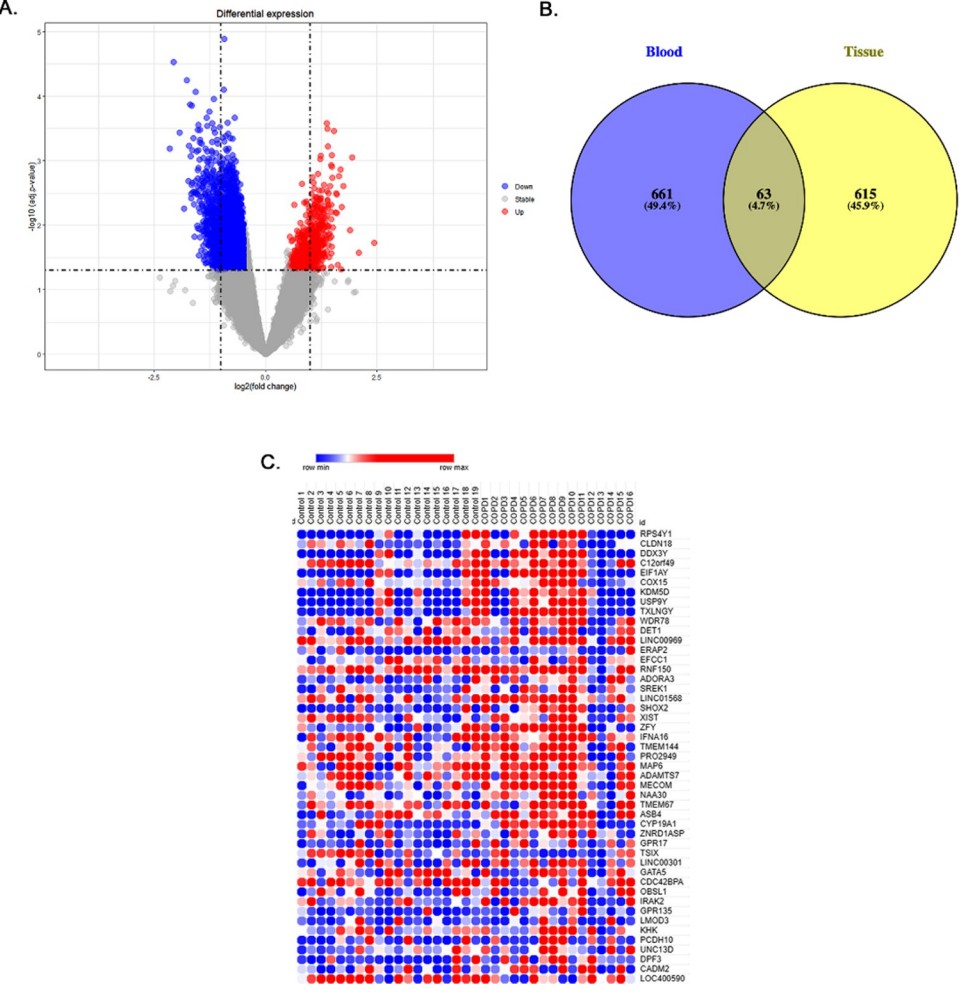

**Fig 1.** B Analysis of COPD differentially expressed genes (DEGs) in comparison to corresponding controls (A) Volcano plots of log fold changes in gene expression. (B) Identification of 63 common DEGs from blood and lung tissue datasets using VENNY. The overlapped area defines the shared DEGs of lung tissue and blood. (C) Heatmap of DEGs with a LogFC > 1.5. Red: up-regulation; green: down-regulation.

### 3.3 GO annotation analysis

Gene Ontology annotation is the process by which functional categories of genes are assigned. The GO annotations of 12 COPD gene clusters showed their enrichment in cell-cell communication, cell regulation, immune processes, transcription factors regulation and ubiquitin pathways. Four of these 12 COPD -gene clusters, *CCDC74B*, *MECOM*, *IRAK2* and *DET1* have shown the lowest FDR values (Table 2), which reflects their highest functional enrichment in i molecular function (MF), biological process (BP), cellular components (CC) categories and KEGG pathways. For the *CCDC74B* cluster, GO enrichment highlights its involvement in 'ubiquitin pathways and protein modification', under the biological processes category, 'Protein Deubiquitination' (GO:0016579) was the top GO term. The other top GO enriched terms falling into remaining categories are as follows; 'proteasome-activating ATPase activity' in MF, 'Proteasome Regulatory Particle' (GO:0005838) in CC and 'Protein degradation' (hsa03050) in KEGG pathways. *IRAK2* cluster was highly involved in signaling pathways and Kinase activity. The GO term in BP highlighted 'Interleukin-1-Mediated Signaling Pathway' (GO:0070498),

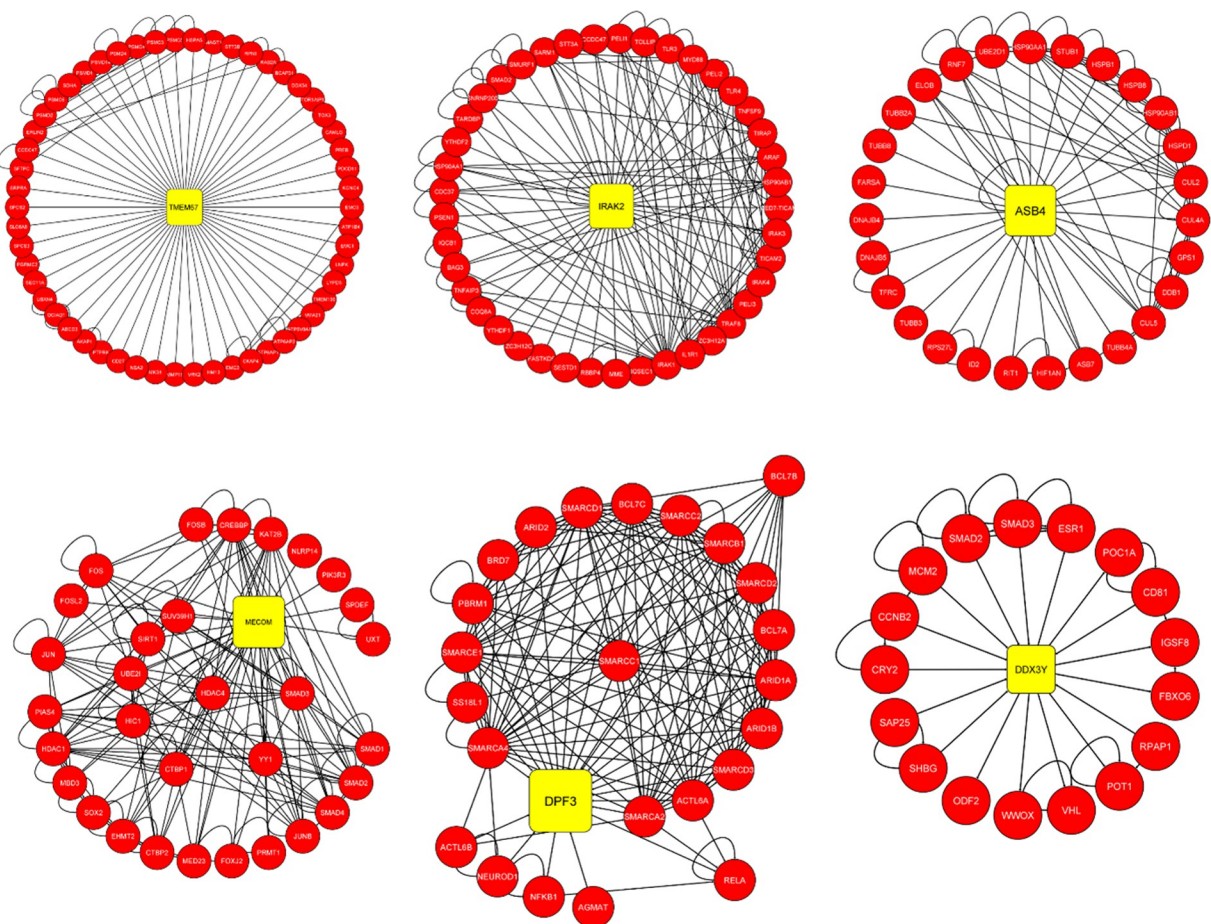

**Fig 2. Hub genes TMEM67, IRAK2, ASB4, MECOM, DPF3 and DDX3Y with their clusters identified from the common DEGs between blood and lung tissue datasets.** Their selection is based on degree of centrality in the PPI network with the score >18.

which mediates cytokine responses during inflammation. The MF ontology source showed 'Protein Kinase Activity' (GO:0004672) and 'Catalytic Activity Acting on A Protein' (GO:0140096) as top GO terms. The CC ontology source was mainly enriched in 'endosome

**Table 1. A total of 12 significant genes with more than 17 of DC were obtained from network analysis and chosen as hub proteins.**

| S.No | Name | Degree | BetweennessCentrality | ClosenessCentrality | Clustering Coefficient |
|------|------|--------|-----------------------|---------------------|------------------------|
| 1 | SREK1 | 77 | 0.004 | 0.456 | 0.177 |
| 2 | TMEM67 | 54 | 0.012 | 0.386 | 0.0405 |
| 3 | IRAK2 | 43 | 0.004 | 0.412 | 0.129 |
| 4 | MECOM | 31 | 0.002 | 0.394 | 0.234 |
| 5 | ASB4 | 29 | $5.15E^{-04}$ | 0.390 | 0.122 |
| 6 | C1QTNF2 | 28 | 0.005 | 0.369 | 0.010 |
| 7 | CDC42BPA | 26 | 0.005 | 0.415 | 0.0289 |
| 8 | DPF3 | 24 | $2.37E^{-04}$ | 0.359 | 0.471 |
| 9 | DET1 | 23 | $3.08E^{-04}$ | 0.384 | 0.260 |
| 10 | CCDC74B | 22 | 0.001 | 0.358 | 0.835 |
| 11 | KHK | 19 | 0.002 | 0.381 | 0.073 |
| 12 | DDX3Y | 17 | $2.80E^{-04}$ | 0.413 | 0.051 |

**Table 2. Functional enrichment of *CCDC74B*, *MECOM*, *IRAK2* and *DET1* clusters, in which highlights highest functional enrichment in different molecular processes like molecular function (MF), biological process (BP), cellular components (CC) and KEGG pathways based on FDR value.**

| DEG Clusters | Ontology | Term ID | Term Description | Observed Gene Count | FDR |
|---|---|---|---|---|---|
| CDC74B | **Biological Process (BP)** | GO:0016579 | Protein Deubiquitination | 20 | 1.64E-31 |
| | | GO:0006511 | Ubiquitin-Dependent Protein Catabolic Process | 19 | 2.42E-25 |
| | | GO:0043687 | Post-Translational Protein Modification | 18 | 3.09E-25 |
| | **Molecular Function (MF)** | GO:0036402 | Proteasome-activating ATPase activity | 6 | 1.28E-13 |
| | | GO:0017025 | TBP-class protein binding | 6 | 5.02E-11 |
| | | GO:0008134 | Transcription factor binding | 8 | 6.89E-06 |
| | **Cellular Component (CC)** | GO:0005838 | Proteasome Regulatory Particle | 19 | 2.98E-48 |
| | | GO:0000502 | Proteasome Complex | 20 | 1.65E-44 |
| | | GO:0031597 | Cytosolic Proteasome Complex | 10 | 2.16E-24 |
| | **KEGG Pathways (KP)** | hsa03050 | Proteasome | 16 | 5.94E-36 |
| | | hsa05169 | Epstein-Barr virus infection | 16 | 9.78E-27 |
| MECOM | **Biological Process (BP)** | GO:0006357 | Regulation of transcription by RNA polymerase II | 29 | 3.62E-19 |
| | | GO:0000122 | Negative regulation of transcription by RNA polymerase II | 21 | 8.33E-19 |
| | | GO:0045892 | Negative regulation of transcription, DNA-templated | 23 | 8.55E-19 |
| | **Molecular Function (MF)** | GO:0043565 | Sequence-specific DNA binding | 22 | 1.03E-18 |
| | | GO:0140110 | Transcription regulator activity | 25 | 7.51E-17 |
| | | GO:1990837 | Sequence-specific double-stranded DNA binding | 18 | 7.02E-16 |
| | **Cellular Component (CC)** | GO:0005654 | Nucleoplasm | 27 | 4.42E-14 |
| | | GO:0031981 | Nuclear Lumen | 28 | 5.76E-14 |
| | | GO:0000785 | Chromatin | 13 | 1.54E-11 |
| | **KEGG Pathways (KP)** | hsa05220 | Chronic myeloid leukemia | 7 | 7.23E-09 |
| | | hsa05200 | Pathways in cancer | 11 | 2.25E-08 |
| | | hsa04068 | FoxO signaling pathway | 7 | 8.41E-08 |
| IRAK2 | **Biological Process (BP)** | GO:0070498 | Interleukin-1-Mediated Signaling Pathway | 11 | 9.63E-16 |
| | | GO:0071347 | Cellular Response To Interleukin-1 | 12 | 6.86E-14 |
| | | GO:0002757 | Immune Response-Activating Signal Transduction | 15 | 1.62E-13 |
| | **Molecular Function (MF)** | GO:0004672 | Protein Kinase Activity | 11 | 2.03E-05 |
| | | GO:0016301 | Kinase Activity | 12 | 2.03E-05 |
| | | GO:0140096 | Catalytic Activity, Acting On A Protein | 17 | 5.41E-05 |
| | **Cellular Component (CC)** | GO:0010008 | Endosome Membrane | 8 | 0.0015 |
| | | GO:0044433 | Cytoplasmic Vesicle Part | 13 | 0.0015 |
| | | GO:0044440 | Endosomal Part | 8 | 0.0015 |
| | **KEGG Pathways (KP)** | hsa04064 | NF-Kappa B Signaling Pathway | 9 | 7.07E-11 |
| | | hsa04620 | Toll-Like Receptor Signaling Pathway | 9 | 7.70E-11 |
| | | hsa05133 | Pertussis | 7 | 1.19E-08 |
| DET1 | **Biological Process (BP)** | GO:0042176 | Regulation Of Protein Catabolic Process | 11 | 1.42E-10 |
| | | GO:0045732 | Positive Regulation Of Protein Catabolic Process | 9 | 1.15E-09 |
| | | GO:1903362 | Regulation Of Cellular Protein Catabolic Process | 9 | 2.44E-09 |
| | **Molecular Function (MF)** | GO:0031625 | Ubiquitin Protein Ligase Binding | 9 | 8.27E-09 |
| | | GO:0048156 | Tau Protein Binding | 3 | 0.0001 |
| | | GO:0004842 | Ubiquitin-Protein Transferase Activity | 6 | 0.00015 |
| | **Cellular Component (CC)** | GO:0080008 | Cul4-RING E3 Ubiquitin Ligase Complex | 7 | 2.55E-12 |
| | | GO:0000151 | Ubiquitin Ligase Complex | 10 | 3.77E-11 |
| | | GO:0031464 | Cul4A-RING E3 Ubiquitin Ligase Complex | 5 | 3.07E-10 |
| | **KEGG Pathways (KP)** | hsa04120 | Ubiquitin Mediated Proteolysis | 9 | 2.32E-12 |
| | | Hh | Nucleotide Excision Repair | 4 | 1.09E-05 |
| | | hsa05215 | Prostate Cancer | 4 | 0.00012 |

membrane' (GO:0010008) and 'Cytoplasmic Vesicle Part' (GO:0044433). KEGG underlined GO terms which are responsible for cytokine production and regulating the immune response like 'NF-Kappa B Signaling Pathway' (hsa04064) and 'Toll-Like Receptor Signaling Pathway' (hsa04620). *DET1* cluster was mostly reported in relation to protein degradation processes. The BP ontology source highlighted 'Regulation of Protein Catabolic Process' (GO:0042176) as the top GO term. MF ontology source identified 'Ubiquitin Protein Ligase Binding' (GO:0031625) as the significant GO term. 'Cul4-RING E3 Ubiquitin Ligase Complex' (GO:0080008) are the top CC terms, while 'Ubiquitin Mediated Proteolysis' (hsa04120) and 'Nucleotide Excision Repair' (hsa03420) was the significant KEGG pathways. (Fig 3A). The MECOM cluster was highly enriched in regulation of transcription by 'RNA polymerase II' as top BP GO term (GO:0006357). Top MF term was 'transcription regulator activity' (GO:0140110). 'Nucleoplasm' (GO:0005654) and 'Nuclear Lumen' (GO:0031981) are the top CC terms, and 'Pathways in cancer' was the significant KEGG pathway (Fig 3B).

The functional enrichment values of the remaining 8 gene clusters (*CDC42BPA*, *DPF3*, *SREK1*, *TMEM67*, *ASB4*, *DDX3Y*, *KHK*, and *C1QTNF2*) are shown in Fig 3A and 3B. *CDC42BPA* cluster predicted its participation in 'Fc Gamma R-Mediated Phagocytosis' (hsa04666). 'Legionellosis' (hsa05134) was mainly enriched by *DPF3* cluster. While the *TMEM67* and *ASB4* clusters were mostly involved in "Proteasome" (hsa03050) and "Ubiquitin-mediated proteolysis" (hsa04120), the *SREK1* cluster was mostly involved in 'RNA binding' (GO:0003723). The 'Pathways in cancer' (hsa05200) was significant GO term in *DDX3Y* cluster. Lastly, *KHK* and *C1QTNF2* were mainly enriched in 'metabolic process' (GO:0044238) and 'extracellular matrix-receptor interaction' (hsa04512) respectively.

### 3.4 Examining the role of hub genes in data from COPD genome wide association studies (GWAS)

Both the GWAS catalog and PhenoScanner databases were used to collate the genetic association data of hub genes with the risk of COPD development. The GWAS data findings of 12 COPD-hub genes, include variant IDs, reference and alternate alleles, significance of association ($P = 5 \times 10^{-8}$), phenotypic traits associated with the query genes, etc. GWAS findings revealed the association of *IRAK2* with eosinophil count alterations usually manifested in inflammatory conditions (Table 3). For the MECOM hub gene, the associated traits are pulmonary complications including COPD, asthma and lung function (Fig 4C and 4D) No significant GWAS data linking the remaining 10 COPD-hub genes (*SREK1*, *TMEM67*, *ASB4*, *C1QTNF2*, *CDC42BPA*, *DPF3*, *DET1*, *CCDC74B*, *KHK*, and *DDX3Y*) with any kind of lung disease was found. In PhenoScanner, 6 out of 12 hub genes (*IRAK2*, *MECOM*, *ASB4*, *CDC42BPA*, *DPF3 and TMEM67*) have revealed an association with lung related traits and lung cancer (S1 Table). *IRAK2* is associated with lung cancer and a high eosinophil count. Genotype-phenotype associations of *MECOM* highlighted pulmonary function interaction, lung cancer, and COPD with acute exacerbation. Both *ASB4* and *CDC42BPA* showed an association with COPD with acute lower respiratory infection. The *DPF3* gene is associated with COPD and squamous cell carcinoma, lung cancer. Lastly, *TMEM67* is associated with lung cancer.

### 3.5 Examining the transcriptional status of COPD hub genes in lung cancer expression

In GPEIA2 analysis, boxplots of 12 hub genes were retrieved. Adenocarcinoma (LUAD) and Squamous Cell Carcinoma (LUSC) were selected with a P-value cutoff of 0.01 using The Cancer Genome Atlas (TCGA) and Genotype-Tissue Expression (GTEx) database. Out of 12 hub

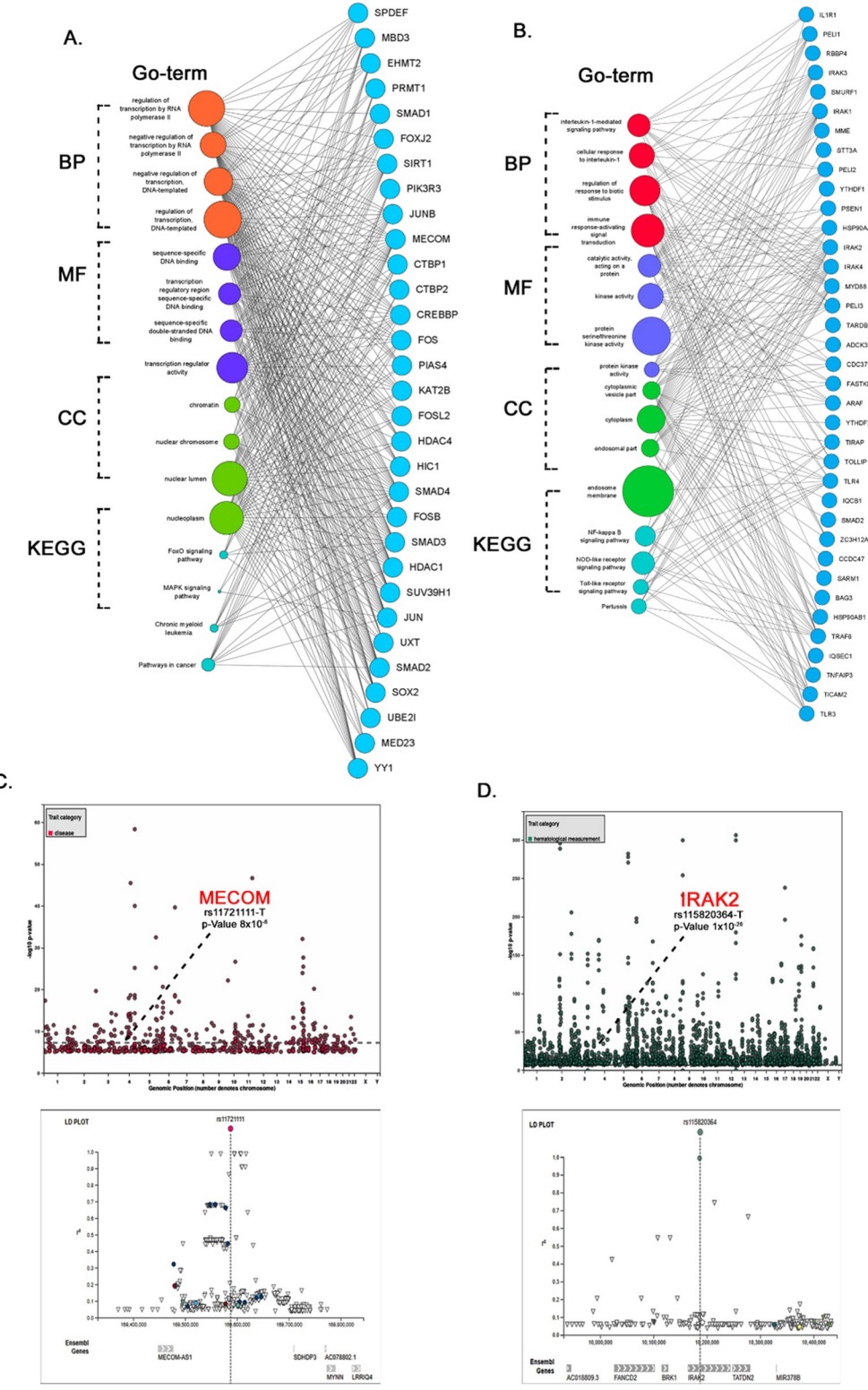

**Fig 3.** GO-annotations stacked network view of (A) *MECOM* and (B) *IRAK2* clusters. The size of the circle (left side) represents the number of genes involved in a specific GO-term. The GWAS loci of (C) *MECOM* (D) *IRAK2* genes from the GWAS catalog.

**Table 3. Association of two hub genes with the lung related traits and lung cancer from GWAS catalog database.**

| Gene | Variant and risk allele | P-value | Reported trait | Trait(s) | Study accession |
|------|------|------|------|------|------|
| IRAK2 | rs114743735 | 6 x 10–11 | Eosinophil percentage of white cells | eosinophil percentage of leukocytes | GCST004600 |
| | rs114743735 | 1 x 10–10 | Eosinophil counts | eosinophil count | GCST004606 |
| | rs115820364- | 1 x 10–26 | Eosinophil counts | eosinophil count | GCST90002298 |
| | rs115820364 | 3 x 10–24 | Eosinophil counts | eosinophil count | GCST90002302 |
| | rs115820364 | 1 x 10–24 | Eosinophil counts | eosinophil count | GCST007065 |
| | rs114743735 | 6 x 10–9 | Sum eosinophil basophil counts | basophil count, eosinophil count | GCST004624 |
| MECOM | rs1344555 | 3 x 10–8 | Pulmonary function | pulmonary function measurement, forced expiratory volume | GCST001251 |
| | rs1344555 | 4 x 10–6 | Pulmonary function (smoking interaction) | pulmonary function measurement, forced expiratory volume, smoking behaviour measurement | GCST001784 |
| | rs11721111 | 8 x 10–6 | Chronic obstructive pulmonary disease | chronic obstructive pulmonary disease | GCST007692 |
| | rs78101726 | 5 x 10–16 | Lung function (FVC) | vital capacity | GCST007429 |
| | rs78101726 | 8 x 10–25 | FEV1 | forced expiratory volume | GCST007432 |
| | rs78101726 | 4 x 10–8 | Lung function (FEV1/FVC) | FEV/FEC ratio | GCST007431 |
| | rs17485347 | 3 x 10–9 | Asthma | asthma | GCST010043 |
| | rs191494905 | 1 x 10–11 | Lung function (FEV1/FVC) | FEV/FEC ratio | GCST007080 |
| | rs6763377 | 9 x 10–10 | Lung function (FEV1/FVC) | FEV/FEC ratio | GCST007080 |
| | rs10936584 | 3 x 10–18 | Lung function (FVC) | vital capacity | GCST007081 |
| | rs6806825 | 5 x 10–12 | Lung function (FVC) | vital capacity | GCST007081 |
| | rs419076 | 2 x 10–24 | Diastolic blood pressure (cigarette smoking interaction) | smoking status measurement, diastolic blood pressure | GCST006187 |
| | rs419076 | 4 x 10–22 | Systolic blood pressure (cigarette smoking interaction) | smoking status measurement, systolic blood pressure | GCST006188 |

genes, only 4 (*IRAK2*, *SREK1*, *C1QTNF2* and *DDX3Y*) have shown significant gene expression in lung cancer compared to the normal tissues (Fig 4A–4E). The boxplots of *IRAK2*, *SREK1* and *C1QTNF2* show their significant expression in LUSCs. The *DDX3Y* gene was significantly expressed in both LAUD and LUSC cells. The GENT2 platform is used to explore gene expression patterns across normal and lung tumor tissues. Fig 5 shows the prognostic value (patient survival in days) of the expression status of 6 COPD-hub genes. Out of the 12-COPD hub genes, 5 genes (*SREK1*, *TMEM67*, *CDC42BPA*, *DPF3*, and *ASB4*) showed an improvement in lung cancer survival duration up to 1500 days (P values for all the associations is <0.02) (Fig 5). The correlation of survival status of patients with different lung cancer subtypes to all five gene expression levels reveals that adenocarcinomas have a longer survival rate (0.2–0.4) than those with squamous and large cell lung cancers.

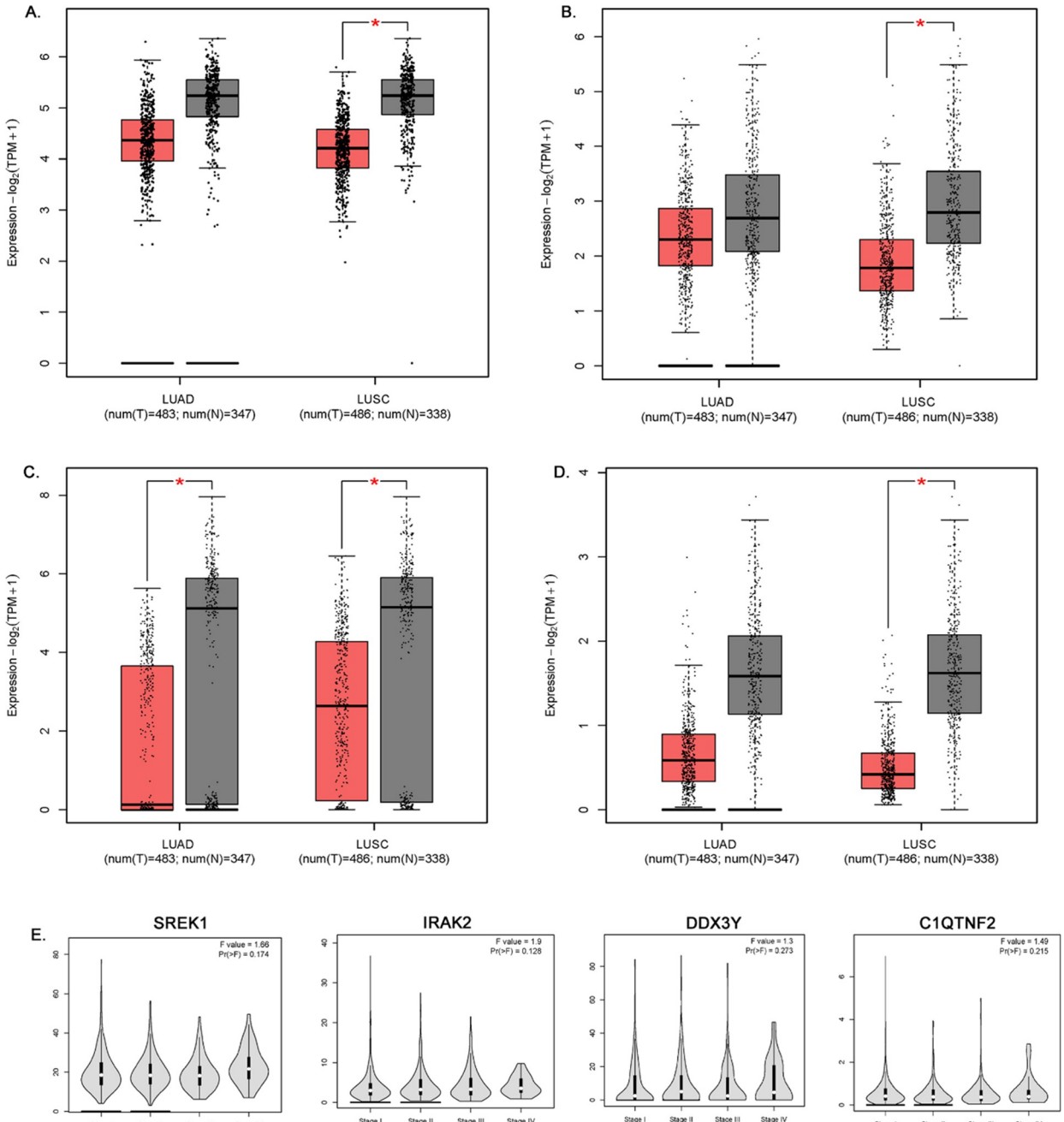

**Fig 4. Expression levels in lung adenocarcinoma and lung squamous cell carcinoma cells in compression to normal tissues from GEPIA2.** A) *SREK1*. B) *IRAK2*. C) *DDX3Y*. D) *C1QTNF2*. The signature score is calculated by mean value of log2 (TPM + 1). The |Log2FC| cutoff of the expression of proposed biomarker was 1. The p-value cutoff of the expression of proposed biomarker was 0.01. The red box indicates the tumor samples while the gray one represents the normal tissues. E. Pathological Stage Plot of *SREK1*, *IRAK2*, *DDX3Y* and *C1QTNF2* genes in lung cancer.

The Human Protein Atlas (HPA) derived protein expression status in normal tissue and lung cancer tissues for COPD- hub genes is illustrated in Fig 6 and Table 4. Abundance of these proteins could be divided into four categories like high, medium, low and not detected by the scoring system based on the intensity of staining, whether strong, moderate, weak, or

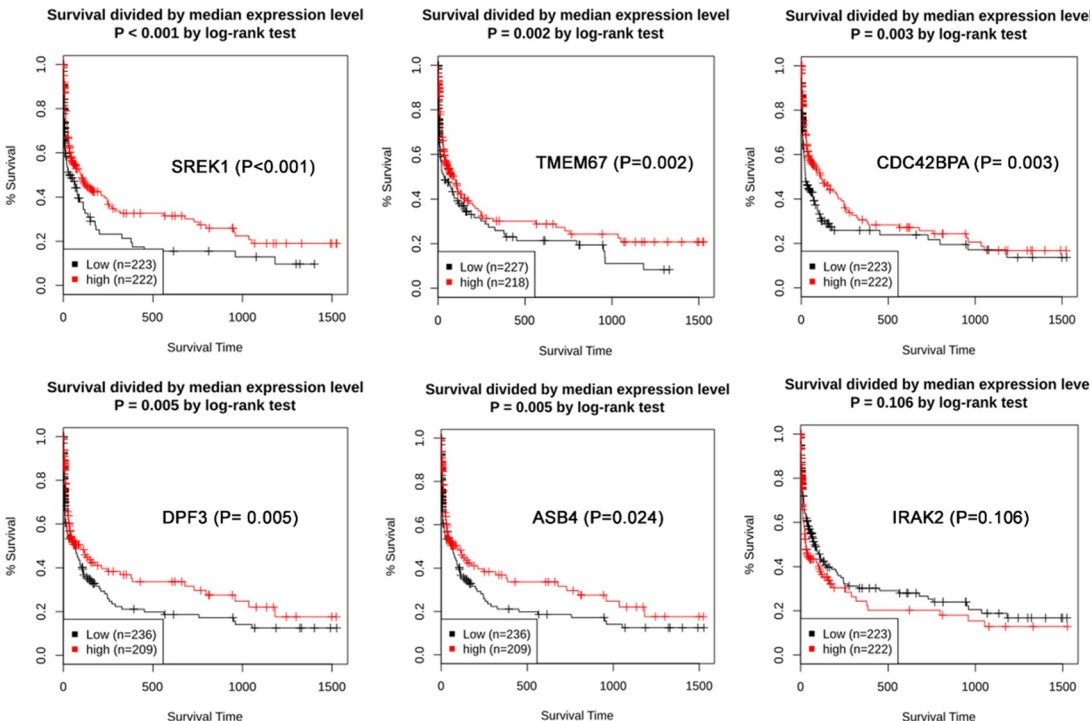

**Fig 5. The prognostic values (patient survival in days) of the expression status of 6 COPD-hub genes.** A) SREK1 (P<0.001).
B) TMEM67 (P = 0.002). C) CDC42BPA (P = 0.003). D) DPF3 (P = 0.005). E) ASB4 (P = 0.024) F) IRAK2 (P = 0106). The
correlation of survival status of patients with different lung cancer subtypes (Squamous, Adeno, Large) to all six genes expression
level.

negative. The macrophage and pneumonocytes staining for ASB4 in normal lung tissues was
not detected and was medium in lung cancer. Medium staining detection of *CDC42BPA* and
*IRAK2* were found in both normal and cancer lung tissue in addition to *SREK1* and *C1QTNF2*
which was observed in medium in normal tissue but higher in lung cancer tissue. Staining of
*MECOM* was very high in both normal and cancer lung tissue. While low to no protein detec-
tion of *DET1* and KHK were observed in both normal and lung cancer tissue. Furthermore,
*DDX3Y* and *DPF3* staining in normal lung tissues were negative but *DDX3Y* was higher in
lung cancer tissue but not DPF3. Finally, data were not available for *TMEM67* and *CCDC74B*
genes in HPA (S3 Fig).

## 4. Discussion

Massive high throughput genome wide- sequencing and expression studies have been effective
in querying the molecular basis of many inherited diseases in humans. However, deciphering
the molecular basis of chronic diseases is challenging, owing to the complex interplay of genes
and environmental factors. The etiopathogenesis of complex diseases like COPD can be better
explained by studying the global gene expression changes. The recent biomarker discoveries in
intracranial aneurysm [25], Parkinson disease [26], Diabetes mellitus [27] and cancers [28]
once again proves the robustness of bioinformatics methods in analyzing the huge gene
expression data. Few studies have attempted to analyze the gene expression changes either in
blood samples [29, 30] or tissue samples of COPD patients [31, 32]. But, none of them
attempted to identify blood based genetic biomarkers. Therefore, this study tried to explore

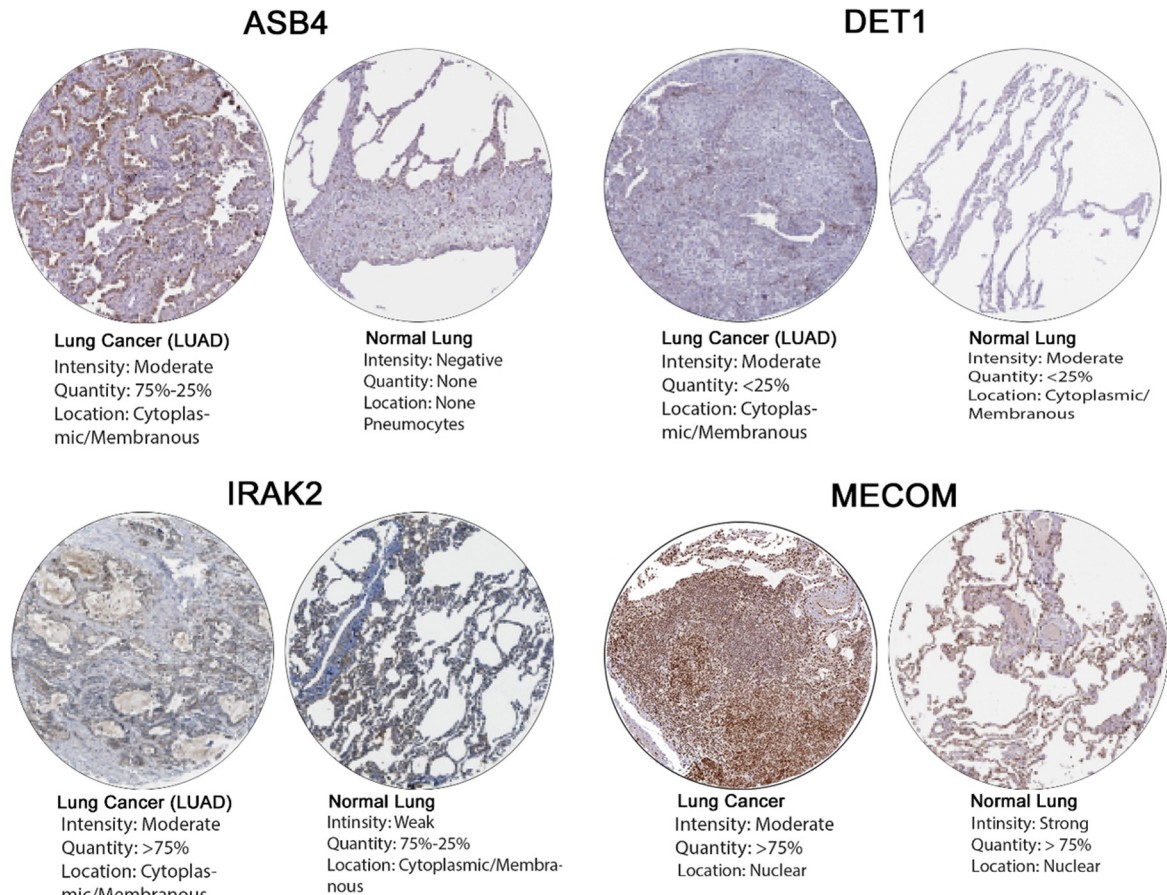

**Fig 6. The expression of the two hub genes from the Human Protein Atlas (HPA) in normal and cancer lung tissue.**

the shared gene expression changes between both blood and lung tissues to identify potential biomarkers to assist in diagnostics or prognostic aspects of COPD patients.

Chronic diseases are caused by the action of dysregulation of multiple genes at different stages of the disease pathology. Hence, we constructed a protein interactome based on the differentially expressed genes in the COPD patients. The protein-protein interaction networks establish the physical contacts between two or more proteins as a result of biochemical events underlying the disease etiopathogenesis. The characteristic features of PPI is based on various connectivity between nodes and edges, where each node indicates a gene connected to its functional partners [23]. To reduce the complexity of PPI network, highest connected nodes are decomposed into a clusters or modules. The gene with the highest number of edges among group of genes within the same cluster is known as hub gene, which are basically chosen based on its degree of centrality (DC) values in the network [33]. DC also refers to node connectivity, i.e. the number of connections to the node and its interaction [34]. In context of these network principles in identifying the COPD-hub genes, we assessed the essential properties of the genes that are involved in the disease.

Since clusters are characterized by extensive connectivity between a set of genes, GO annotations provides functional interpretation of them under vaiety of biological categories [35]. In the present study we identified 12 COPD hub gene clusters (SREK1, TMEM67, IRAK2, MECOM, ASB4, C1QTNF2, CDC42BPA, DPF3, DET1, CCDC74B, KHK, DDX3Y) from PPI

Table 4. The expression levels of the 10 hub genes in normal lung and cancer tissues: Human Protein Atlas (HPA).

| Genes | Normal Tissue Staining | | | Cancer Lung Tissue (Tumor cell) | | |
|---|---|---|---|---|---|---|
| | Cell | Staining | Quantity | Staining | Quantity | Type of Cancer |
| ASB4 | Macrophage | Not detected | None | Medium | 75%-25% | LUAD |
| | Pneumonocyte | Not detected | None | | | |
| CDC42BPA | Macrophage | Medium | >75% | Medium | 75%-25% | LUSC |
| | Pneumonocyte | Medium | >75% | | | |
| DET1 | Macrophage | Low | <25% | Low | <25% | LUSC |
| | Pneumonocyte | Not Detected | None | | | |
| IRAK2 | Macrophage | Medium | >75% | Medium | >75% | LUSC |
| | Pneumonocyte | Low | 75%-25% | | | |
| MECOM | Macrophage | High | >75% | High | >75% | LUSC |
| | Pneumonocyte | High | >75% | | | |
| DDX3Y | Macrophage | Not detected | None | High | 75%-25% | LUSC |
| | Pneumonocyte | Not detected | None | | | |
| SREK1 | Macrophage | Medium | 75%-25% | High | >75% | LUSC |
| | Pneumonocyte | Medium | 75%-25% | | | |
| DPF3 | Macrophage | Not Detected | <25% | Not Detected | None | LUAD |
| | Pneumonocyte | Not detected | <25% | | | |
| C1QTNF2 | Macrophage | Not detected | None | High | >75% | LAUD |
| | Pneumonocyte | Medium | 75%-25% | | | |
| KHK | Macrophage | Medium | >75% | Low | >75% | LAUD |
| | Macrophage | Not detected | None | | | |

network, which revealed their enrichment in cell regulation, immune process, transcription factors regulations and protein degradation pathways. The upregulated gene (DDX3Y) in both blood and lung tissue were enriched in functions associated with regulation of gene expression, cell cycle, cellular senescence and FoxO signaling pathway which is involved in many cellular physiological events such as apoptosis and cell-cycle control. Moreover, there were two down-regulated genes (MECOM and KHK). MECOM were associated with regulation of transcription, pathways in cancers and FoxO signaling pathway. While KHK were involved in Starch and sucrose metabolic processes. However, of those 12 gene clusters, CCDC74B, MECOM, IRAK2 and DET1 clusters had shown the lowest FDR values, which reflects their highest functional enrichment in different molecular processes. The CCDC74B gene cluster was mainly enriched in proteasome pathway, which degrades unneeded proteins within the cell. The activity of proteasome can be impaired by cigarette smoke resulting in reduction of antigen presentation and lead to prolonged lung infections and COPD patients [36].

In lung tissues of COPD patients, accumulation of ubiquitylated proteins and further degradation by proteasome machinery is reported [37]. Protein catabolic processes pathway enriched in DET1 gene cluster also plays an important role in pathogenesis of COPD. The chronic inflammation in COPD contributes to the imbalance of protein degradation resulting in the loss of skeletal muscle protein, one of the characteristic features present in COPD [38]. On the other hand, MECOM gene cluster highlights the regulation of transcriptional pathway which controls the changes in gene transcription of many inflammatory substances that play a key role in the pathogenesis of COPD [39, 40]. The IRAK2 gene cluster showed its involvement in regulation of inflammatory process such as interleukin (IL)-1 pathway activation and Toll like receptor that is directly linked to the pathogenesis of COPD, is characterized by abnormal release of inflammatory cytokines, remodeling of the airways and dysregulated immune cell activity [41, 42].

Genome wide association studies reveals the association of genetic variants with risk of developing common diseases by screening genetic samples from thousands of samples. In this study, 12 hub genes were searched in GWAS databases for their association with COPD, lung function traits as well as lung cancers. The GWAS data confirmed that the variants in the 6 COPD-hub genes (IRAK2, MECOM, CDC42BPA, ASB4, DPF3 and TMEM67) shows genome wide significant association to traits that could potentially modify the risk of COPD pathology development. At least 5 variants in IRAK2 were significantly associated with variety of eosinophil count traits [43]. Eosinophilia (high eosinophil counts) causes inflammation of the lung tissue and exacerbates the lung function in the COPD patients. However, the role of eosinophils in COPD is unclear, as not all COPD patients develop eosinophilic airway inflammation [44, 45]. Interestingly, IL-1 signaling has been shown to be associated with eosinophilic inflammatory profiles in patient with COPD [43]. Moreover, in COPD patients with eosinophilic inflammation have the tendency to respond to steroid therapy. Therefore, eosinophil count is an important point of view to direct biological therapies for COPD [46]. Many variants in MECOM were strongly associated with FEV1and other traits that are directly related to lung function and COPD pathogenesis [47]. Other COPD- hub genes (CDC42BPA, ASB4, DPF3 and TMEM67) are were also associated with lung function related traits and lung cancer [48–50].

COPD is one of the significant risk factors for oncogenesis of the lung tissues, which is seen in about 1% of COPD patients every year [51]. Both COPD and lung cancer share many common pathways such as immune dysfunction and regulation of transcription factors [52]. Interestingly, pathways enriched by MECOM and IRAK2 were involved in lung cancer development. For instance, MECOM is an important transcription factor involved in oncogenesis [53, 54]. Aberrant expression of MECOM is one of the characteristic features of many malignancies including leukemia [55] and solid tumors such as breast cancer and hepatocellular carcinoma [53, 56] as well as lung cancer [57]. Moreover, frequent alterations in MECOM have been associated with primary and metastatic lung adenocarcinomas [58]. On the other hand, activation of the TLR pathway has a significant impact on cancer progression regulation including lung cancer [59, 60]. One genetic variant in IRAK2 (rs779901 C > T) in the TLR signaling pathway is suggested to be a prognostic biomarker for non-small cell lung cancer (NSCLC) [61]. Global gene expression profile analysis provides a valuable insight into the normal biological process and to disease pathogenesis [62]. To support the contribution of IRAK2 and MECOM hub genes, significant dysregulation of expression in lung cancer types were observed in HPA, GPEIA2 and GENT2 databases as well. Furthermore, differentially expression of IRAK2 and MECOM genes has been reported in many studies in cancers or COPD [63–65].

## 5. Conclusions

In conclusion, we identified, 12 blood based molecular biomarkers (SREK1, TMEM67, IRAK2, MECOM, ASB4, C1QTNF2, CDC42BPA, DPF3, DET1, CCDC74B, KHK, DDX3Y) for COPD diagnosis, by integrative gene expression and gene network approaches. Out of these 12 hub genes, two (MECOM and IRAK2) were over expressed in lung cancers tissues, which reflects a shared molecular lineage between COPD and lung cancers. Interestingly, we have also identified that the expression status of other COPD hub genes like SREK1, TMEM67, CDC42BPA, DPF3, and ASB4 improves the survival duration of lung cancer patients, hence they may act as potential molecular drug targets and/or biomarkers for both COPD and/or lung cancer. However, biological and clinical relevance of each COPD hub gene can be better understood, when our findings are explored through future *in vitro* and *in vivo* validation assays.

## Supporting information

**S1 Fig. Overview of PPI network constructed from 63 common genes using cytoscape STRING database.** The PPI network at p-value >0.05 consist of 995 nodes interact with 18924 edges.
(PDF)

**S2 Fig. The 12 hub gene protein interaction network.** (A) Cluster-1 (SREK1). (B) Cluster-2 (TMEM67). (C) Cluster-3 (IRAK2). (D) Cluster-4 (MECOM). (E) Cluster-5 (ASB4). (F) Cluster-6 (CDC42BPA). (G) Cluster-7 (DPF3). (H) Cluster-8 (DET1). (I) Cluster-9 (KHK). (J) Cluster-10 (DDX3Y), the hub gene selected based on high centrality in the protein network of DEGs.
(PDF)

**S3 Fig. Histopathological images of DEGS.** Protein Pathology Atlas of 12 hug genes in normal lung and lung cancer tissues.
(PDF)

**S1 Table. Hub genes GWAS association.** Genetic association of hub genes with the lung related traits and lung cancer from Phenoscanner database.
(DOCX)

## Author Contributions

**Conceptualization:** Babajan Banaganapalli, Ramu Elango, Noor A. Shaik.

**Data curation:** Babajan Banaganapalli, Bayan Mallah, Kawthar Saad Alghamdi.

**Formal analysis:** Babajan Banaganapalli, Bayan Mallah, Kawthar Saad Alghamdi.

**Funding acquisition:** Babajan Banaganapalli.

**Investigation:** Babajan Banaganapalli, Noor A. Shaik.

**Methodology:** Babajan Banaganapalli, Bayan Mallah, Kawthar Saad Alghamdi.

**Project administration:** Babajan Banaganapalli.

**Resources:** Babajan Banaganapalli.

**Software:** Babajan Banaganapalli.

**Supervision:** Babajan Banaganapalli, Ramu Elango, Noor A. Shaik.

**Validation:** Babajan Banaganapalli, Bayan Mallah, Kawthar Saad Alghamdi.

**Visualization:** Babajan Banaganapalli, Bayan Mallah.

**Writing – original draft:** Babajan Banaganapalli, Bayan Mallah, Noor A. Shaik.

**Writing – review & editing:** Babajan Banaganapalli, Walaa F. Albaqami, Dalal Sameer Alshaer, Nuha Alrayes, Ramu Elango, Noor A. Shaik.

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
