## [Decision Letter · Decision Letter 0]

1 May 2022

PONE-D-22-07850Integrative Weighted Molecular Network Construction from Transcriptomics and Genome Wide Association Data Identified Shared Genetic Biomarkers for COPD and Lung CancerPLOS ONE

Dear Dr. Banaganapalli,

Thank you for submitting your manuscript to PLOS ONE. After careful consideration, we feel that it has merit but does not fully meet PLOS ONE’s publication criteria as it currently stands. Therefore, we invite you to submit a revised version of the manuscript that addresses the points raised during the review process.

We look forward to receiving your revised manuscript.

Kind regards,

Narasimha Reddy Parine, Ph.D

Academic Editor

PLOS ONE

Journal Requirements:

"This project was funded by the Deanship of Scientific Research (DSR) at King Abdulaziz University, under Grant no. G-593-140-1441. The authors, therefore, acknowledge the DSR for technical and financial support."

We note that you have provided funding information. However, funding information should not appear in the Acknowledgments section or other areas of your manuscript. We will only publish funding information present in the Funding Statement section of the online submission form. 

"Funding Author: BB

Grant Number: G-593-140-1441

Full Name of Funder: Deanship of Scientific Research (DSR), King Abdulaziz University

URL: https://dsr.kau.edu.sa/

Role: The funders had no role in

study design, data collection and analysis,

decision to publish, or preparation of the

manuscript."

Reviewers' comments:

Reviewer's Responses to Questions

**Comments to the Author**

1. Is the manuscript technically sound, and do the data support the conclusions?

Reviewer #1: Yes

Reviewer #2: Yes

2. Has the statistical analysis been performed appropriately and rigorously? 

Reviewer #1: Yes

Reviewer #2: Yes

3. Have the authors made all data underlying the findings in their manuscript fully available?

Reviewer #1: Yes

Reviewer #2: Yes

4. Is the manuscript presented in an intelligible fashion and written in standard English?

Reviewer #1: Yes

Reviewer #2: Yes

5. Review Comments to the Author

Reviewer #1: The original article by Banaganapalli and colleagues sought to identify significant changes in COPD and lung cancer. The authors collected transcriptomics and protein-protein interaction networks, and then discovered some differentially expressed genes, that are analyzed by some bioinformatics approaches and are available in silico datasets. They further predicted these clinicopathological events and patient outcomes through the features they selected. Other comments are listed as follows

• There are different stages of COPD to determine the severity of disease status, is there any difference among their gene expression?

• Some figures have poor pixels and the font is too small.

• They did not describe the statistical algorithm and p-value of each website and panel.

• The number of DEGs in manuscript and figure 1B are different.

• Some of the web links they provide cannot be entered

• There is no title of Table 4. It is also better to use scoring system to quantify and present the IHC results of these genes.

Reviewer #2: In the present study, the authors aim to identify potential biomarkers for COPD diagnosis and prognosis by analyzing the dysregulated gene expression patterns from blood and lung tissues. They identified many common DEGs from blood and lung tissue datasets and focused on 12 hub genes with DC score of >17. They further performed PPI analysis, GO analysis and GWAS to reveal the significance of 12 hub genes in COPD and lung cancer. They provided massive and valuable information about COPD and lung cancer.

Minor Points:

1. There are many references without formal styles such as Ref #10, #12, #13, #20, #28 #31, #39 and The Ref#16 is wrongly embedded. The authors should correct them.

2. The refractive asthma should be corrected as refractory asthma.

3. There are many inconsistencies among figures, legends and contexts such as Figure 1 & 3 and table

4. 12 significant genes should be with more than 17 of DC in the legends of Table 1 and Figure 2.

5. Go-annotations should be corrected as GO-annotations in Figure 3.

6. The items in table 1 should be re-modify to a professional one and the style in the column of Term Description should be consistent.

7. The image and staining intensity of MECOM from HPA database compared to Figure S3 were wrongly embedded and the authors should correct them.

6. PLOS authors have the option to publish the peer review history of their article (what does this mean?). If published, this will include your full peer review and any attached files.

Reviewer #1: No

Reviewer #2: **Yes: **Magesh Ramasamy

---

## [Author Response · Author response to Decision Letter 0]

16 Jun 2022

Reviewer #1: The original article by Banaganapalli and colleagues sought to identify significant changes in COPD and lung cancer. The authors collected transcriptomics and protein-protein interaction networks, and then discovered some differentially expressed genes, that are analyzed by some bioinformatics approaches and are available in silico datasets. They further predicted these clinicopathological events and patient outcomes through the features they selected. Other comments are listed as follows

• There are different stages of COPD to determine the severity of disease status, is there any difference among their gene expression?

Answer: Our study aimed to identify the common biomarker between COPD and lung cancer. For this reason, we ignored the disease severity variability in our study. In this study, we collected all stages of COPD (GSE54837).

• Some figures have poor pixels, and the font is too small.

Answer: We increased the quality of figures 

• They did not describe the statistical algorithm and p-value of each website and panel.

Answer: WE revised as per the reviewer suggestion

• The number of DEGs in manuscript and figure 1B are different.

Answer: Corrected 

• Some of the web links they provide cannot be entered

Answer: Weblinks updated 

• There is no title of Table 4. It is also better to use scoring system to quantify and present the IHC results of these genes.

Answer: Updated in the table 4

Reviewer #2: In the present study, the authors aim to identify potential biomarkers for COPD diagnosis and prognosis by analyzing the dysregulated gene expression patterns from blood and lung tissues. They identified many common DEGs from blood and lung tissue datasets and focused on 12 hub genes with DC score of >17. They further performed PPI analysis, GO analysis and GWAS to reveal the significance of 12 hub genes in COPD and lung cancer. They provided massive and valuable information about COPD and lung cancer.

Minor Points:

1. There are many references without formal styles such as Ref #10, #12, #13, #20, #28 #31, #39 and The Ref#16 is wrongly embedded. The authors should correct them.

Answer: All the references have been updated with reference manger software 

2. The refractive asthma should be corrected as refractory asthma.

Answer: Corrected

3. There are many inconsistencies among figures, legends and contexts such as Figure 1 & 3 and table

Answer: Updated 

4. 12 significant genes should be with more than 17 of DC in the legends of Table 1 and Figure 2.

Answer: Table 1 shows the degree centrality of 12 significant genes, whereas Figure 2 mentioned >18 centrality score genes

5. Go-annotations should be corrected as GO-annotations in Figure 3.

Answer: corrected 

6. The items in table 1 should be modified to a professional one and the style in the column of Term Description should be consistent.

Answer: Table 1 is centrality table, all the values are consistence according to the journal style 

7. The image and staining intensity of MECOM from HPA database compared to Figure S3 were wrongly embedded and the authors should correct them.

Answer: Figure 6 was revised and now its exactly mapping to Figure S3

---

## [Decision Letter · Decision Letter 1]

1 Sep 2022

Integrative Weighted Molecular Network Construction from Transcriptomics and Genome Wide Association Data Identified Shared Genetic Biomarkers for COPD and Lung Cancer

PONE-D-22-07850R1

Dear Dr. Banaganapalli,

We’re pleased to inform you that your manuscript has been judged scientifically suitable for publication and will be formally accepted for publication once it meets all outstanding technical requirements.

Kind regards,

Narasimha Reddy Parine, Ph.D

Academic Editor

PLOS ONE

Reviewers' comments:

Reviewer's Responses to Questions

**Comments to the Author**

1. If the authors have adequately addressed your comments raised in a previous round of review and you feel that this manuscript is now acceptable for publication, you may indicate that here to bypass the “Comments to the Author” section, enter your conflict of interest statement in the “Confidential to Editor” section, and submit your "Accept" recommendation.

Reviewer #1: All comments have been addressed

Reviewer #3: All comments have been addressed

2. Is the manuscript technically sound, and do the data support the conclusions?

Reviewer #1: Yes

Reviewer #3: Yes

3. Has the statistical analysis been performed appropriately and rigorously? 

Reviewer #1: Yes

Reviewer #3: Yes

4. Have the authors made all data underlying the findings in their manuscript fully available?

Reviewer #1: Yes

Reviewer #3: Yes

5. Is the manuscript presented in an intelligible fashion and written in standard English?

Reviewer #1: Yes

Reviewer #3: Yes

6. Review Comments to the Author

Reviewer #1: (No Response)

Reviewer #3: In this study, the authors analyzed the gene expression differences between the blood and lung tissues of COPD patients to identify diagnostic and prognostic biomarkers. They have performed multiple computational biology analysis methods (differential gene expression analysis, gene networking, hub gene identification, GO annotations, and druggability analysis) to discover the genetic biomarkers. They have also queried the candidate hub genes in GWAS and in cancer expression databases to understand their contribution to lung cancer. The manuscript is well written. The number of figures and tables is sufficient. I am convinced that the new information identified in COPD may lead to developing future treatment strategies. Therefore, I sincerely recommend this article for publication in your journal.

7. PLOS authors have the option to publish the peer review history of their article (what does this mean?). If published, this will include your full peer review and any attached files.

Reviewer #1: No

Reviewer #3: No

---

## [Editor Report · Acceptance letter]

26 Sep 2022

PONE-D-22-07850R1 

Integrative Weighted Molecular Network Construction from Transcriptomics and Genome Wide Association Data to Identify Shared Genetic Biomarkers for COPD and Lung Cancer 

Dear Dr. Banaganapalli:

I'm pleased to inform you that your manuscript has been deemed suitable for publication in PLOS ONE. Congratulations! Your manuscript is now with our production department. 

Kind regards, 

on behalf of

Dr. Narasimha Reddy Parine 

Academic Editor

PLOS ONE